# Characterization of Phages YuuY, KaiHaiDragon, and OneinaGillian Isolated from *Microbacterium foliorum*

**DOI:** 10.3390/ijms23126609

**Published:** 2022-06-14

**Authors:** Uylae Kim, Elizabeth S. Paul, Arturo Diaz

**Affiliations:** Biology Department, La Sierra University, Riverside, CA 92505, USA; ukim761@lasierra.edu (U.K.); epau753@lasierra.edu (E.S.P.)

**Keywords:** bacteriophages, phages, bioinformatics, genomics, host range, phage therapy

## Abstract

*Microbacterium foliorum* is a Gram-positive bacteria found in organic matter. Three lytic bacteriophages, KaiHaiDragon, OneinaGillian, and YuuY, were isolated from *M. foliorum* strain NRRL B-24224. Phage YuuY in particular expresses a broad host range as it possesses the ability to infect closely related bacterial species *Microbacterium aerolatum* at a high plating efficiency. Characterization tests were performed on all three *Microbacterium* phage to assess morphology, genomic characteristics, pH and thermal stabilities, life cycle, and the type of receptor used for infection. All three phages showed similar pH stability, ranging from pH 5–11, except for KaiHaiDragon, which had a reduced infection effectiveness at a pH of 11. YuuY possessed a significantly higher temperature tolerance compared to the other *Microbacterium* phages as some phage particles remained viable after incubation temperatures of up to 80 °C. Based on the one-step growth curve assay, all three *Microbacterium* phages possessed a relatively short latent period of 90 min and an approximately two-fold burst size factor. Moreover, all three phages utilize a carbohydrate receptor to initiate infection. Based on bioinformatics analysis, YuuY, KaiHaiDragon and OneinaGillian were assigned to clusters EA10, EC, and EG, respectively.

## 1. Introduction

*Microbacterium* spp. consist of yellow, rod-shaped, Gram-positive bacteria that are commonly found in plants, soil, water, and dairy products [1]. *Microbacterium foliorum*, for example, is an aerobic Actinobacteria that is often isolated from the phyllosphere of grasses, ripened cheese, vegetables, and fruit [1,2]. Due to their prevalence in the outdoors, various strains of *Microbacterium*, including *M. foliorum* and *M. paraoxydans*, have been isolated from wound swabs, blood samples, and other clinical specimens [3,4]. In response to cases of *Microbacterium*-induced bacteremia surrounding catheters of immunocompromised patients, the need to explore biomedical interventions that can be used to regulate specific populations of bacteria arises. One potential solution is phage therapy. Despite the importance of bacteriophages in the targeted lysis of specific pathogenic bacteria, the potential of a wide host range or capability of infecting multiple species is yet to be harnessed. With over 50 species of *Microbacterium*, the heterogeneity of this genera and their prevalence in both the environment and clinical setting make bacteriophages that are capable of infecting multiple strains of *Microbacterium* useful and efficient [3]. Furthermore, identifying novel bacteriophages that can lyse pathogenic *Microbacterium* strains that are found in fresh produce and dairy products can help control contaminations during the pre- and post-harvest stages of food production, and ultimately minimize food borne illnesses [5]. 

Contributions made from basic phage research expand our understanding about virus–host cell relationships, modes of infection, host specificity, and viral structure, ultimately leading to the advancement of biotechnology that can help humanity against deadly outbreaks caused by pathogenic bacteria [6]. In order to contribute to the viral diversity of *Microbacterium*-infecting bacteriophages, three phages are isolated from environment samples collected in Riverside, California using *Microbacterium foliorum* as the host bacteria. Phages identified and sequenced through the Science Education Alliance-Phage Hunters Advancing Genomics and Evolutionary Science (SEA-PHAGES) program are assigned to clusters based on genomic similarity (www.phagesbd.org (accessed on 9 June 2022)). The threshold value adopted for cluster inclusion is nucleotide sequence similarity spanning 50% genome lengths, which is then adjusted to 35% shared gene content [7,8]. This classification system provides a clearer way of organizing and understanding the evolutionary relationships between newly characterized phages while providing insight into the shared morphological and genomic characteristics of phage within a specific cluster or subcluster [8]. All three phages were lytic and viral particle morphology and stability at various environmental conditions was determined. Moreover, the genomes of all three phages were sequenced and annotated. Lytic *Siphoviridae* phages KaiHaiDragon and OneinaGillian were assigned to Cluster EC and Cluster EG, respectively, based on their genomic structure. Notably, the end of the OneinaGillian genome had an assembly gene order that was highly dissimilar to the rest of the Cluster EG phages [9]. Phage YuuY was assigned to a small subcluster, EA10, which comprises only four other lytic *Siphoviridae* bacteriophages. YuuY is the only phage in its subcluster that possesses a tRNA. Interestingly, aside from infecting *M. foliorum*, results from host range assays showed that phage YuuY was capable of infecting *M. aerolatum*. 

## 2. Results

### 2.1. General Characteristics of Microbacterium Phages

Phage particles were isolated from moist soil samples in Riverside County using either a direct or enriched isolation approach before being plated directly onto bacterial lawns containing the *M. foliorum* host. Phages KaiHaiDragon and OneinaGillian were isolated using the direct isolation method, whereas YuuY was isolated from an enriched culture. After a 48-h incubation period, individual plaques were selected for two rounds of purification and amplification into high-titer phage lysates. Electron micrographs revealed that all three of the isolated and purified bacteriophages possessed morphological features that were characteristic of the Siphoviridae family, such as a non-enveloped head, a non-contractile tail, and double-stranded DNA (Figure 1A–C). All three bacteriophages possessed lytic life cycles, producing clear circular plaques that were around 2 mm in diameter (Figure 1D–F) while genetic analysis showed an absence of integrase and repressor genes. The absence of lysogeny-related genes is a key characteristic of lytic *Siphoviridae* phages [10].

### 2.2. Determination of Host Range

In order to determine the host-specificity of each phage, or the ability to infect different *Microbacterium* strains, each phage was tested against *M. aerolatum* strain NRRL B-24228 and *M. paraoxydans* strain NRRL B-24275, as *M. aerolatum* and *M. paraoxydans* share a 98.53% and 98.5% 16S rRNA gene homology, respectively, with *M. foliorum* [3]. Since the bacteriophages were originally isolated using *Microbacterium foliorum* strain NRRL B-24224, all three phages were able to mount infections against *M. foliorum* at a high efficiency of plating, yielding titers of at least 5 × 10^9^ pfu/mL. When testing the phages against alternative *Microbacterium* strains such as *M. aerolatum* and *M. paraoxydans*, however, the efficiency of plating (EOP) varied significantly for each phage (Table 1). Phages KaiHaiDragon and OneinaGillian were only able to infect *M. aerolatum* at high lysate concentrations (10^−1^ dilution), which are likely false positives because of the lysis of bacterial cells without phage infection [11]. In contrast, phage YuuY was able to mount an infection across lower lysate concentrations with a high EOP, indicating that phage YuuY is capable of infecting another bacterial host, *M. aerolatum*. None of the phages were able to successfully infect *M. paraoxydans*. 

### 2.3. Determination of pH Stability 

The stability of phages in relation to environmental factors such as temperature and pH should be considered when using phages as antibacterial agents. In order to assess the pH stability of each phage, 1 × 10^6^ viral particles were suspended in PYCa growth browths of varying acidities for 60 min, mixed with M. foliorum, and then incubated for 48 h before comparing the titers. Phages OneinaGillian and YuuY were able to maintain a high infection rate from a pH of 5 to 11 (Figure 2). For phage OneinaGillian, the infectious units (PFU/mL) at a pH of 7 versus a pH of 9 or 11 were not significant (pH 7 vs. 9, *p*-value = 0.075485; pH 7 vs. 11, *p*-value = 0.093226). Phage KaiHaiDragon possessed a slightly narrower pH stability than the other two Microbacterium phages as infection capability decreased significantly at a pH of 11 compared to the normal broth pH of 7 (*p*-value = 0.021695) (Figure 2). YuuY was unable to infect *M. foliorum* at a pH of 3, whereas infection for phages KaiHaiDragon and OneinaGillian were reduced at this pH by 98% compared to their normal infection rates at pH of 7 (Figure 2). 

### 2.4. Determination of Thermal Stability

The thermal stability of each phage was determined by measuring infection capability after incubating 1 × 10^7^ viral particles at six different temperatures (30 °C, 37 °C, 50 °C, 60 °C, 70 °C, and 80 °C) for 30 min before mixing them with M. foliorum. All three bacteriophages were able to replicate and produce comparable titers when the phage particles were incubated between 30 °C (normal incubation temperature) and 60 °C (Figure 3). While phages KaiHaiDragon and OneinaGillian were not able to mount infections at all after being heated 70 °C or higher, phage YuuY was able to mount infections after both 70 °C and 80 °C incubation periods at 15% and 2% effectiveness compared to 30 °C, respectively (Figure 3). 

### 2.5. Determination of Latent Period and Burst Size

The latent period, which is the period between the adsorption of the phage to the host bacterium and the beginning of the lysis of the host bacterium, was approximately 90 min for all three phages. The average burst sizes for KaiHaiDragon, OneinaGillian, and YuuY were 48, 69, and 88 PFU per bacterial cell, respectively (Figure 4). 

### 2.6. Determination of Receptor Type

The adsorption of the phage to the bacterial surface is the first and most important step in the phage infection process. Peptidoglycans and teichoic acids, important components of the cell wall of Gram-positive bacteria, are often involved in bacteriophage adsorption [12]. Peptidoglycans are polymers composed of multiple units of amino acids and sugar derivatives, while teichoic acids are polysaccharides composed of glycerol phosphate or ribitol phosphate and amino acids. To determine whether the type of bacterial receptors utilized by each phage to induce infection are predominantly made of proteins or sugar derivatives, host strain *M. foliorum* NRRL B-24224 was treated with either proteinase K or a combination of sodium acetate and sodium periodate in order to degrade the proteins or carbohydrates, respectively, present on the surface of the bacterial cells. Incubation of KaiHaiDragon, OneinaGillian, and YuuY with sodium periodate prevented phage binding, whereas incubation in proteinase K and sodium acetate alone did not (Figure 5). 

### 2.7. Genomic Characteristics of Microbacterium Phages

The genome lengths, GC% content, termini, and lytic life cycles of KaiHaiDragon, OneinaGillian, and YuuY were used to categorize them into Cluster EC, Cluster EG, and Subcluster EA10, respectively (Table 2). OneinaGillian has direct terminal repeats while KaiHaiDragon and YuuY have circularly permuted termini, consistent with headful packaging systems. Phages OneinaGillian and YuuY have each half of their genome transcribed in different directions, while phage KaiHaiDragon has all of its open reading frames (ORF) transcribed in the same direction (Figure 6). 

Cluster EC phage KaiHaiDragon possesses a genome of 52,992 bp, with a G+C content of 68.9% and a circularly permeated (Cir Per) termini (Table 2). Only around one-third of KaiHaiDragon’s 91 genes have assigned functions (30). When comparing KaiHaiDragon to other EC phages, it was found to be 91.04% and 76.97% similar to ClearAsMud and Megan, respectively (Figure 6). KaiHaiDragon gp83 encodes for an RNA polymerase sigma factor that is found in Cluster EC and Cluster EI phages. The sigma factor is likely involved in controlling phage gene expression (Figure 6). There is also an 18 bp asymmetric sequence motif, 5′-TAgaCTATagGTgTaAgC-3′ (capital letters signify complete conservation and lower-case letters are those appearing in at least 9 of the 12 instances), which is repeated 12 times throughout the genome, positioned around 21–30 bp upstream of a predicted translation initiation codon and upstream of the putative ribosome binding sequence (Figure 6). 

Cluster EG phage OneinaGillian has a longer genome length (61,703 bp) than KaiHaiDragon and YuuY, as the average size of the 23 Cluster EG phages is 62,222 bp. OneinaGillian also has 203-bp direct terminal repeats (DTR) and a G+C content of 67.1% (Table 2). When compared to Cluster EG phages Hyperion and Squash, OneinaGillian’s genome had a 70.49% and 72.11% similarity, respectively (Figure 7). For reference, the genomes of Squash and Hyperion are 90.75% similar. An 18 bp conserved short inverted repeat motif, 5′-GATCAACCtGggttgatc-3′ (capital letters signify complete conservation and lower case letters indicate the presence of single nucleotide variation) is found in 10 intergenic regions in OneinaGillian (Figure 7). This same inverted repeat motif can be found towards the right end of the genomes of all three Cluster EG phages, but are spaced out differently for OneinaGillian [14]. It should be noted that the repeats are conserved in OneinaGillian despite significant differences in gene content compared to Hyperion and Squash. When comparing the shared homologies of the Cluster EG phages by pairwise alignment, OneinaGillian has large regions of dissimilarity at the left and right hand of the genome in comparison to Hyperion and Squash (Figure 7). 

Phage YuuY possesses a genome of 40,996 bp, with a G+C content of 63.2% and a circularly permuted (Cir Per) termini (Table 2). The Subcluster EA10 differs from other Cluster EA phages because of their lysis cassette, such as a distinct gene that encodes lysin A (YuuY gp 23), membrane proteins (located between YuuY gp23 and gp31), and genes with no known function (NKF) located downstream of the thymidylate synthase (YuuY gp 48) towards the end of the genome (Figure 8). YuuY shared 86.77% and 86.89% nucleotide identity with Subcluster EA10 phages Nucci and Quartz, respectively, which were 97.81% similar to each other (Figure 8). Unlike phages KaiHaiDragon and OneinaGillian, YuuY actually encodes for one alanine tRNA, with complementary anticodon CGC (start at bp 29,310 and stop at bp 29241). YuuY encodes for a Cas4-like exonuclease, a RecA-like DNA recombinase, and the lysis cassette includes an endolysin and a holin and is downstream of the tail measure protein and tail proteins. All Cluster EA10 phages lack a capsid maturation protease gene directly downstream of the portal protein gene, a trait that is common in most Cluster EA phages. 

## 3. Discussion

Bacteriophages are host-specific viruses that infect bacteria and as such they could have distinct applications, such as the control of harmful or unwanted bacterial strains. The universal decline in the effectiveness of antibiotics has generated renewed interest in the use of phages to treat bacterial infections, either as an alternative or a supplement to antibiotic treatment [15]. Aside from potential clinical uses, having a better understanding of virus–host interactions, host specificity, how phages infect bacteria and the mechanisms used by bacteria to resist infection have led to the advancement of virology in general as well as various applications in biotechnology [6].

*Microbacterium* sp. mainly comprises aerobic Gram-positive bacteria with high G+C content and a peptidoglycan with B-type cross linkage [16]. *Microbacterium* species have been isolated from food, plants and soil [1]. Additionally, there has been an emergence of clinical cases associated with *Microbacterium* spp., including bacteremia in patients [4] and in a cystic fibrosis patient [17]. Moreover, *Microbacterium* strains have potential application in biotechnology [18] and nitrogen fixation [19]. Thus, efforts to characterize the abundant phage population that is yet to be isolated from the soil [20] helps us prepare for a surge of novel diseases caused by potentially pathogenic bacteria. 

Here, we report on the isolation and characterization of three *Microbacterium* phages belonging to the *Siphoviridae* family. All three phages formed large clear plaques and bioinformatic analysis showed lack of integrase genes (Figure 6, Figure 7 and Figure 8), agreeing with previous reports that *Microbacterium* phages are mostly lytic [14]. Additionally, one-step growth curve assays showed that all three phages have a latent period of 90 min and a burst size factor of approximately two-fold, indicating that all three phages could be potential candidates for phage therapy.

The potential use of bacteriophages in either phage therapy or other biotech applications requires the knowledge of the stability of the phage particles under different environmental conditions as well as the host range. Temperature plays an important role in the stability of the phage particles and capacity for attachment [21]. All three phages were stable between 30 °C and 60 °C, with KaiHaiDragon and OneinaGillian being inactivated at temperatures above 70 °C, which is in agreement with the results from other phages work in which biological activity of the phages was significantly lower at 55 °C and completely inactivated at 65 °C [22,23]. Studies have shown that high temperatures inactivate phages due to nucleic acid and protein denaturation [24]. Interestingly, phage YuuY retains some infectivity after being incubated at 70 °C and even at 80 °C, although there was a 6-fold and 50-fold reduction, respectively, compared to that at 30 °C. It should also be noted that the phage particles retained their viability for at least six months after being stored at 4 °C (data not shown). Further experiments will need to be conducted to explore YuuY’s thermal stability. 

Acidity and alkalinity are other important factors that influence phage stability. Phage particles were inactivated at pH 3 or below for all three phages but remained relatively stable at a pH range of 5 to 11. The optimum pH was 7 and YuuY phage particles maintained the most consistent activity at pH 5, 9, and 11. This is important, as phage particles used for phage therapy will likely be exposed to factors that may cause their inactivation in the human body, such as contact with the acidic gastric pH and alkaline bile salts [25]. Moreover, the strong stability of these bacteriophages could prove useful for a number of applications in the food industry. For example, the stability of phages during heat pasteurization [26] and cooking during industrial food production [27] is particularly important in their application as an additive. 

Host range is a key property for phage therapy as well as the biology of bacteriophages in general. Unlike KaiHaiDragon and OneinaGillian, YuuY was able to infect *M. aerolatum* at a relatively high efficiency, making it a more suitable candidate for phage therapy. For phages that infect Gram-positive bacteria, polysaccharides on the surface of the bacteria and peptidoglycans have been reported as phage receptors [12]. An assay to allow for an initial prediction of the nature of the host receptor, protein or carbohydrate, concluded that all three phages likely use a carbohydrate receptor to mediate adsorption. Since YuuY was able to infect *M. aerolatum*, it suggests that it uses a different carbohydrate receptor compared to that of KaiHaiDragon and OneinaGillian. Xia et al. demonstrated that phages that infect *Staphylococcus aureus*, another Gram-positive bacteria, attach to the N-acetyl glucosamine moiety of teichoic acids [28]. Likewise, *Bacillus subtilis* phages SP2 and SP10 bind to the D-glucose chain of teichoic acid [29]. Further experiments will have to be conducted to identify the specific receptor(s) bound by KaiHaiDragon, OneinaGillian, and YuuY. 

One general mode of virus adaptation involves the acquisition of host genes. Viral genomes commonly encode homologs of genes that are involved in host metabolism [30]. Most analyzed dsDNA phages encode components of the DNA replication machinery to avoid relying exclusively on regulation by host replication enzymes [31]. Our bioinformatics analysis revealed the presence of a gene analogous to an RNA polymerase sigma factor in phage YuuY. Interestingly, an 18 bp asymmetric sequence motif is repeated 12 times throughout the genome, five of which are found among the structural genes in the left part of the genome while seven are located among the non-structural genes in the right part of the genome. The repeated motifs are found upstream of the predictive putative ribosome binding sequence and 21–30 bp upstream of predicted gene start sites. For bacterial σ-factors, such as σ^70^ in *Escherichia coli*, consensus sequence elements located around positions −35 and −10 (with respect to the transcription start site at +1) confer promoter specificity on the RNA polymerase [32]. However, at this point, it is unclear whether the phage encoded RNA sigma factor homolog binds to the motifs and whether these motifs are involved in regulating phage gene expression or another aspect of lytic growth. 

In summary, we report the isolation and characterization of three new lytic phages that infect *M. foliorum*. Phage YuuY is of particular interest as it has desirable characteristics for a therapeutic phage, including strong lytic activity and no genes indicative of a lysogenic lifestyle, phage particles that are stable at various pH and temperature ranges, and the ability to infect more than one *Microbacterium* species. 

## 4. Materials and Methods

### 4.1. Microbacterium Strains and Host Range Assay

The bacterial strain used to isolate all three phages was *M. foliorum* NRRL B-24224. The following Microbacterium strains were obtained from the American Research Service Culture Collection—Northern Regional Research Laboratory (NRRL) repository: *M. aerolatum* NRRL B-24228 and *M. paraoxydans* NRRL B-24275. Bacteria were grown using PYCa media (containing per 1 L volume: 1.0 g yeast extract, 15 g peptone, 2.5 mL 40% dextrose, and 4.5 mL 1 M CaCl_2_) in approximately 48-h intervals on a rotary shaker set at 30 °C, 250 rpm. The average doubling time for *M. foliorum* was 1.35 h (sd = 0.85 h, n = 62 samples).

For the host range assay, bacterial lawns of each strain of Microbacterium were used for spot titer assays [33]. Phage lysate dilutions down to 10^−8^ were prepared and plated using the double-agar method. The efficiency of plating (EOP) was calculated by dividing the titer of phage on the alternative host by the titer of phage on the original *M. foliorum* host [33]. The EOP was reported as “1/number of phage particles plated” if the phage failed to infect an alternative *Microbacterium* strain [33]. Host range assays were performed three times for each phage. 

### 4.2. Isolation and Sequencing of Microbacterium Bacteriophages

All three phages were isolated from soil samples collected in Riverside, California by students in La Sierra University’s SEA-PHAGES program (seaphages.org (accessed on 3 June 2022). Soil samples were treated with phage buffer (10 mM Tris-HCL, pH 7.5; 10 mM MgSO_4_; 68.5 mM NaCl; 1 mM CaCl_2_), shaken vigorously for 2 h, filtered, and plated directly on solid overlays containing 0.35% agar and *M. foliorum* host and incubated at 30 °C for 36–48 h. If no plaques were visible, 0.25 mL of *M. foliorum* was added to the filtered phage lysate and incubated at 30 °C for 48–72 h to generate an enriched culture. Individual plaques were purified twice and high-titer phage lysates were generated before DNA was purified using the Wizard DNA clean-up kit (A7280; Promega) [33]. Sequencing libraries were prepared at the Pittsburgh Bacteriophage Institute using an NEB Ultra II FS DNA library kit. These libraries were run on an Illumina MiSeq instrument to yield base reads for each phage, and raw reads were constructed into a single contig for each phage by means of Newbler and Consed assembly, providing 1039-fold, 2049-fold, and 2327-fold shotgun coverage for KaiHaiDragon (accession number MH590600), OneinaGillian (accession number MH727556), and YuuY (accession number MN284901), respectively. 

### 4.3. Electron Microscopy

Firstly, 5 × 10^9^ phage particles were spotted onto formvar and carbon-coated 400 mesh copper grids, then they were rinsed with distilled water and stained with 1% uranyl acetate. Samples were imaged using a Talos 120 TEM microscope. 

### 4.4. Bioinformatics 

Genome annotations were carried out manually using the Phage Evidence Collection and Annotation Network (PECAAN) (discover.kbrinsgd.org accessed on 9 June 2022), which includes Starterator, Glimmer [34], and GeneMark [35] software to help identify transcriptional initiation sites. The coding capacity and function of phage genes was assigned based on results obtained from PhagesDB [36] and NCBI BLAST [37], HHPred [38], the Conserved Domain Database [39], TMHMM [40], TOPCONS [41], and tRNAscan-SE [42]. Whole genome sequences of bacteriophages were visualized, as well as comparatively analyzed using Phamerator [13]. Dot plots of phages KaiHaiDragon, OneinaGillian, and YuuY against other phages in their respective clusters were created by running the phages’ FASTA sequences through Gepard [43]. Percent identities between phages of the same cluster were calculated using NCBI BLAST. 

### 4.5. pH and Thermal Stability Assays

In order to determine the pH range in which the three Microbacterium phage particles could remain viable, 1 × 10^6^ viral particles were suspended in PYCa media at different pH levels (3, 5, 7, 9, 11). For each experimental condition, 5 μL of diluted phage lysate was aliquoted into 495 μL of PYCa buffer, and left for an hour to incubate at 30 °C. Phage lysates were then added to 250 μL of the host bacteria *M. foliorum* and plated after a 15 min incubation using the double-agar-layer method [44]. Titers were calculated after a 48-h incubation period at 30 °C. 

The thermal stability range for each bacteriophage was assessed by suspending 1 × 10^7^ viral particles at 30 °C, 37 °C, 50 °C, 60 °C, 70 °C, or 80 °C. For each experimental condition, 5 μL of diluted phage lysate was aliquoted into 495 μL of PYCa buffer and left to incubate for 30 min at each set temperature. Tubes containing phage lysates were then placed on ice for 3 min before being added to host bacteria and plated using the double-agar-layer method. Titer was calculated after a 48-h incubation period at 30 °C. Both pH and thermal stability assays were performed at least three times for each phage. 

### 4.6. One-Step Growth Curve

One-step growth curve protocols were modified from Ellis and Delbruck [45] in order to determine the latent period and burst size associated with the lytic life cycle of *Microbacterium foliorum* infecting phages. Log-phase *M. foliorum* culture was collected at OD_600_ ≈ 0.250 (cells concentration ≈ 4 × 10^6^ cfu/mL). The culture was centrifuged for 1000× *g* for 10 min before 0.1 mL of diluted phage (1 × 10^7^ pfu/mL) was added to 0.9 mL of concentrated cells. After a 3 min incubation period at 30 °C and a third centrifugation for 1 min at 13,500× *g*, all free phage particles were discarded and the cell pellet was resuspended in 1 mL of pre-warmed 30 °C PYCa medium. Phage growth was measured by conducting plaque assays every 15 min for a duration of 240 min using the double-agar-layer method and calculating the titers after a 48-h incubation period at 30 °C. Early time points that remained stagnant in plaque-forming units were averaged to determine the Time = 0 titer, and all later time point titers were divided by this value. The beginning and end of the sigmoidal curve were used to calculate the latent period and burst size, respectively. One-step growth curve experiments were conducted at least three times for each phage. 

### 4.7. Receptor Assay

*M. foliorum* bacteria was harvested at an OD_600_ of 1.00–1.25 and treated with either proteinase K (final concentration of 0.2 mg/mL), 1.5 mL of 50 mM sodium acetate (pH 5.5), or 1.5 mL of 50 mM sodium acetate in combination with 1.5 mL of 100 mM sodium periodate. For proteinase K treatment, treated cells were incubated at 37 °C for a 3-hour period. For the sodium acetate and combo treatments, tubes containing treated cells were covered with aluminum foil and incubated at room temperature for 2 h. Treatment with sodium acetate alone without sodium periodate served as the negative control. Incubation periods were followed by three washes with PYCa media, then the addition of 1 × 10^6^ pfu/mL of phage for 15 min to allow for phage absorption prior to high-speed centrifugation (13,000× *g*). Particles that failed to infect the bacteria were transferred to initiate a second round of infection on untreated bacterial cells. Thus, if the phage receptor is not available to the phage, most of the particles should remain in the supernatant, resulting in a higher titer in the second infection. Phage-infected cultures for each type of treatment were plated using the double-agar-layer technique and the titers were calculated after a 48-h incubation period at 30 °C. Receptor assays were conducted at least three times for each phage. 

### 4.8. Statistical Analysis

Two-tailed *t*-tests were conducted for the pH and thermal stability assays to calculate significant differences in viral replication in optimum conditions (pH 7 and 30 °C, respectively) versus the various test conditions. Differences were considered significant at *p*-value < 0.05. This same statistical test was carried out for the receptor assay to compare the control group with either the proteinase K treatment group or the sodium periodate combo treatment group.

## Figures and Tables

**Figure 1 ijms-23-06609-f001:**
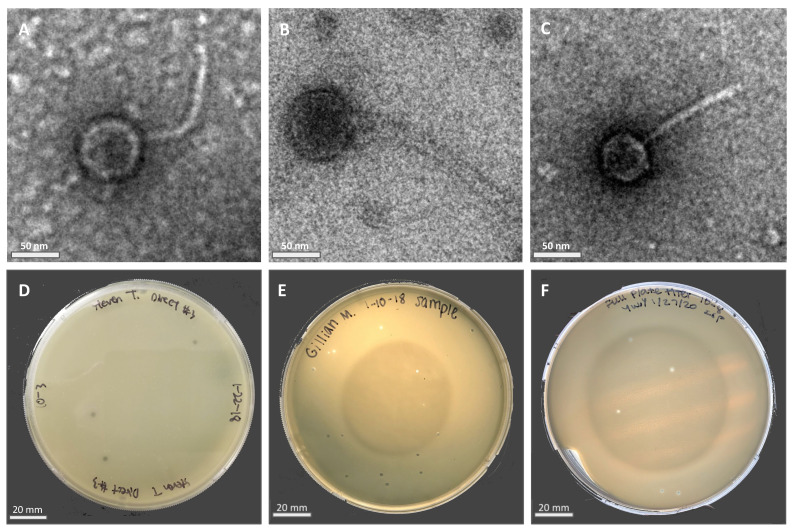
Electron micrographs (EM) of negatively stained *Microbacterium* phages with their respective plaque morphologies: (**A**) KaiHaiDragon; (**B**) OneinaGillian; (**C**) YuuY. Plaques of phages plated on *M. foliorum* lawns: (**D**) KaiHaiDragon; (**E**) OneinaGillian; (**F**) YuuY.

**Figure 2 ijms-23-06609-f002:**
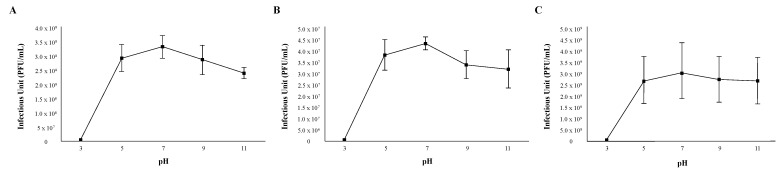
pH stability of phages. Error bars depict the standard deviations between the titers generated from three independent assays: (**A**) KaiHaiDragon; (**B**) OneinaGillian; (**C**) YuuY.

**Figure 3 ijms-23-06609-f003:**
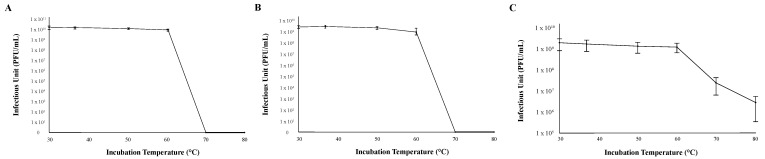
Thermal stability of phages. Error bars depict the standard deviations between the titers generated from three independent assays: (**A**) KaiHaiDragon; (**B**) OneinaGillian; (**C**) YuuY.

**Figure 4 ijms-23-06609-f004:**
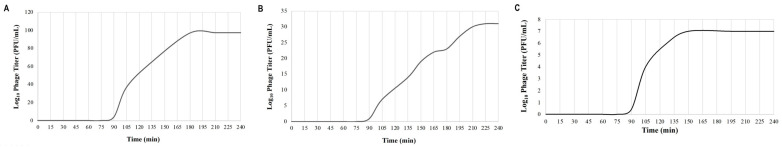
One-step growth curve assays depicting the life cycle of each phage on isolated host strain *M. foliorum* NRRL B-24224 at 30 °C for a 4 h duration: (**A**) KaiHaiDragon; (**B**) OneinaGillian; (**C**) YuuY.

**Figure 5 ijms-23-06609-f005:**
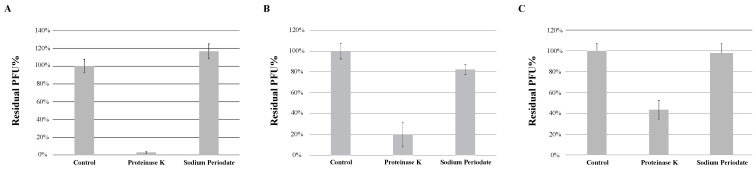
Effects of different treatments of bacteria on phage absorption, shown as residual PFU percentages. Effect of proteinase K treatment and sodium periodate treatment on the absorption of (**A**) KaiHaiDragon, (**B**) OneinaGillian, and (**C**) YuuY to *M. foliorum*. Error bars depict the standard deviations between the titers generated from three independent assays.

**Figure 6 ijms-23-06609-f006:**
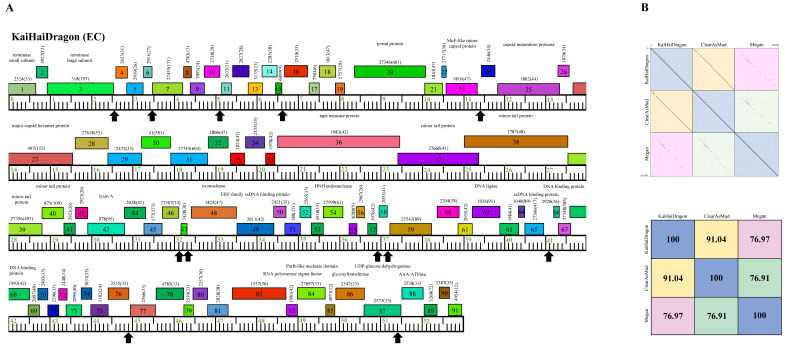
Genomic organization of *Microbacterium* Cluster EC phage KaiHaiDragon. (**A**) Whole genome organization of phage KaiHaiDragon. Genes are represented as boxes above the genome, reflecting rightwards-transcription. Genes are colored according to their phamily designations using Phamerator [13]. The phamily number, along with the number of members in the specific phamily in parentheses, are also written above the boxes. Black up arrows indicate the positions of conserved 18 bp asymmetric sequence motifs. (**B**) Dot plot and percent identity comparisons of KaiHaiDragon to two other Cluster EC phages, ClearAsMud and Megan.

**Figure 7 ijms-23-06609-f007:**
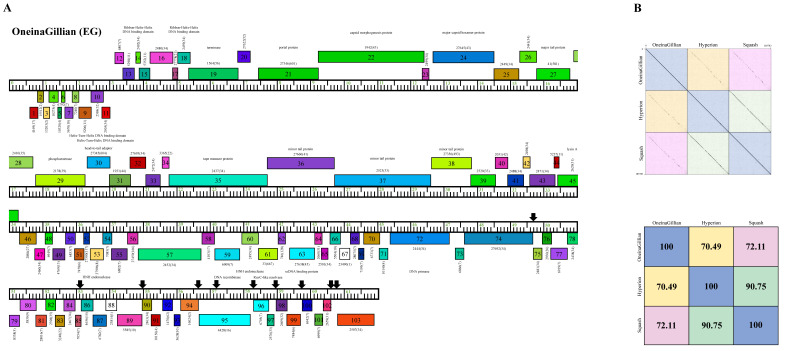
Genomic organization of *Microbacterium* Cluster EG phage OneinaGillian. (**A**) Whole genome organization of phage OneinaGillian. Genes are represented as boxes above or below the genome, reflecting rightwards or leftwards transcription, respectively. The phamily number, along with the number of members in the specific phamily in parentheses, are also written above or below the boxes. Black down arrows indicate the positions of a conserved 18 bp short inverted repeat motif. (**B**) Dot plot and percent identity comparisons of OneinaGillian to two other Cluster EG phages, Hyperion and Squash.

**Figure 8 ijms-23-06609-f008:**
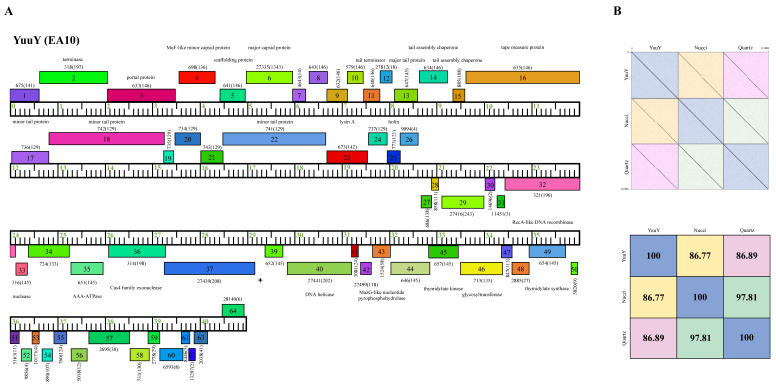
Genomic organization of *Microbacterium* Subcluster EA10 phage YuuY. (**A**) Whole genome organization of phage YuuY. Genes are represented as boxes above or below the genome, reflecting rightwards or leftwards-transcription, respectively. The phamily number, along with the number of members in the specific phamily in parentheses, are also written above or below the boxes. (**B**) Dot plot and percent identity comparisons of YuuY to two other Subcluster EA10 phages, Nucci and Quartz.

**Table 1 ijms-23-06609-t001:** Efficiency of plating (EOP) values of each phage on various *Microbacterium* strains relative to *M. foliorum*.

Phage	Cluster	Isolation Host	*M. foliorum*	*M. aerolatum*	*M. paraoxydans*
KaiHaiDragon	EC	*Microbacterium foliorum* NRRL B-24224	1	~5 × 10^−8^	<1/(~7 × 10^8^)
OneinaGillian	EG	*Microbacterium foliorum* NRRL B-24224	1	~1 × 10^−5^	<1/(~3 × 10^9^)
YuuY	EA	*Microbacterium foliorum* NRRL B-24224	1	~2 × 10^−3^	<1/(~2 × 10^10^)

**Table 2 ijms-23-06609-t002:** Genometrics of *Microbacterium* phages.

Phage	Cluster	Isolation Host	Length (bp)	GC%	Terminal	Morphology	tRNA	Life Cycle	Accession #
KaiHaiDragon	EC	*Microbacterium foliorum* NRRL B-24224	52,992	68.9%	Cir Per	*Siphoviridae*	0	Lytic	MH590600
OneinaGillian	EG	*Microbacterium foliorum* NRRL B-24224	61,703	67.1%	203 bp DR	*Siphoviridae*	0	Lytic	MH727556
YuuY	EA	*Microbacterium foliorum* NRRL B-24224	40,996	63.2%	Cir Per	*Siphoviridae*	1	Lytic	MN284901

## Data Availability

The nucleotide sequences of the phages are available in Genbank under accession numbers as indicated in the Materials and Methods.

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
