# Peer review of "Characterization of Phages YuuY, KaiHaiDragon, and OneinaGillian Isolated from Microbacterium foliorum"

_ijms, 2022, doi:10.3390/ijms23126609_

Round 1

Reviewer 1 Report

The manuscript by Kim et al describes the isolation and characterization of three phages isolated against Microbacterium foliorum from soil samples near Riverside, California. Some species in Micobacterium genera are known to be opportunistic pathogens, hence isolation and subsequent study of any phages against Microbacterium spp carries future potential clinical relevance. The authors isolated, sequenced, and studied three phages. All three are sipho-phages with strictly lytic lifestyle. Of the three one phage (YuuY) infects at least one other closely related species. Through genomic characterization and analysis, the three phages are grouped into three different clusters, EC, EG, and EA10 clusters. Among the genome features, the authors report the presence of sequence motifs that are repeated several times in two of the genomes and the presence of a single t-RNA gene (alanyl-tRNA) in YuuY phage. Dot plot analyses shows how related these phages are to other members in the respective clusters.

The authors also determine other characteristics of these phages such as burst size, latent period, environmental stability and make an attempt to narrow the host receptor type.

Overall, this is a well-written and scientifically sound manuscript with good relevance to the field. I think this manuscript can be further strengthened by making some changes to how the data and text are presented. Also, the discussion section needs to include more of a discussion of the features of these phages not a mere repetition of findings of this paper.

Specific comments:

Introduction section needs to include a background about the different clusters with relevant references and why is this clustering important in the context of the current study.

All images in the figures are blurry. Please use high resolution images.

Figure 1A-C: Why are the images of different phages at different magnification level? It is hard to compare the three phages when they are shown at different magnification levels.

Line 89: What does “Higher EOP” mean? Does it mean higher than the EOP of phage KaiHaiDragon?

Table 1: The phage titers are reported to 2 significant digits but I see not standard deviation. Please approximate it to the nearest whole number and place ~ in front of the number.

Line 95: Please include a rationale for doing pH  and thermal stability studies.

Line 97: M. smegmatis???

Section 2.6: How do you know that the bacteria are alive after various treatments? The apparent lack of adsorption might be because the cells are dead.

Discussion: Please include in your discussion the implications of the repeating sequence motifs in the genome and relatedness of the phages in this study to previously characterized phages in the cluster.  

Reviewer 2 Report

Dear Authors,

The work here presented characterises three novel bacteriophages. After reading throughout the manuscript, some major issues need to be corrected for clarity and consistence of the results reported. Please read carefully my comments below and be sure to add to the manuscript a reference and/or the accession number for the genome of the phage YuuY (otherwise the data availability is not respected) and a paragraph about the statistical analysis.

Here in detail my comments:

L 46-58  References for these statements are missing

L 69-71                  The presence of clear circular plaques alone is not an indication of lytic phages. The analysis of the genome should support this statement. Please verify and correct in the manuscript.

L 314-320             References to the methods are needed if the method is not described thoroughly

L324       Is there a website for the SEA-PHAGES programme? Please include the link.

L 322-336             Is the genome of the phage YuuY published too? It is important to provide the reference or the accession number, otherwise your genomic analysis is not verificable 

L 329-330             Please describe or include a reference to the purification and phage-enrichment protocol used

L114       How long was the phage incubated at the different temperatures before mixing them with M. foliorum? Please include in the text.

L 125      The sentence belongs to the Materials and Methods section

L 137-153             This information is not relevant in the Result section and it contains a lot of methodology. If needed, move the information to the appropriate section.

L 156-157             This sentence belongs to the Discussion section

L 164-168             Too much repetition of Methods, please reduce and get straight to the results.

L 180      It’s Figure 6, not 7.

L 181-182             This belongs to Discussion

L 182-185             If possible, use the IUPAC codes for degenerate DNA, rather than upper/lower case (if the information about the nucleotide variation is known)

Legends of Figure 6, 7 and 8        “phamily” is an incorrect term. Please re-write the concept in scientific terminology

L 257-259             Here is what I was talking about in L 69-71. Include this result in the Results section.

L 267-269             Please give some details about the other works cited here.

L 274-275             “Dissect” can lead to misinterpretation, please rephrase the sentence.

L 276-280             This part is only repetition of results, there is no comparison with other studies, no discussion. Please rewrite. Furthermore, specify in what terms the “bacteriophages could prove useful for a number of applications in the food industry”.

L 281-289             Again, no discussion here, only repetition of results. Please rewrite

L 374 and 376     Please report the centrifugation speed in “g” rather than “rpm”

In Materials and Methods there is no mention to any statistics, which are reported in the Results. Please add a paragraph describing the statistical analysis undertaken during the work.

L 413-414             In the Materials and Methods there are no accession numbers to the phage genomes.

Round 2

Reviewer 1 Report

My comments have been addressed by the authors.

Reviewer 2 Report

Dear Authors,

Thank you for applying the suggested reviews to your manuscript.

The clarity of the presentation has improved drastically and, from my perspective, I have no further suggestions.

Kind Regards